# Non-equilibrium dissipative supramolecular materials with a tunable lifetime

Marta Tena-Solsona[1,2,*], Benedikt Rieß[1,*], Raphael K. Grötsch[1], Franziska C. Löhrer[3], Caren Wanzke[1], Benjamin Käsdorf[4], Andreas R. Bausch[5], Peter Müller-Buschbaum[3], Oliver Lieleg[4] & Job Boekhoven[1,2]

Many biological materials exist in non-equilibrium states driven by the irreversible consumption of high-energy molecules like ATP or GTP. These energy-dissipating structures are governed by kinetics and are thus endowed with unique properties including spatio-temporal control over their presence. Here we show man-made equivalents of materials driven by the consumption of high-energy molecules and explore their unique properties. A chemical reaction network converts dicarboxylates into metastable anhydrides driven by the irreversible consumption of carbodiimide fuels. The anhydrides hydrolyse rapidly to the original dicarboxylates and are designed to assemble into hydrophobic colloids, hydrogels or inks. The spatiotemporal control over the formation and degradation of materials allows for the development of colloids that release hydrophobic contents in a predictable fashion, temporary self-erasing inks and transient hydrogels. Moreover, we show that each material can be re-used for several cycles.

[1] Department of Chemistry, Technische Universität München, Lichtenbergstrasse 4, 85748 Garching, Germany. [2] Institute for Advanced Study, Technische Universität München, Lichtenbergstrasse 2a, 85748 Garching, Germany. [3] Lehrstuhl für Funktionelle Materialien, Physik-Department, Technische Universität München, James-Franck-Straße 1, 85748 Garching, Germany. [4] Department of Mechanical Engineering and Munich School of Bioengineering, Technische Universität München, Boltzmannstrasse 11, Garching 85748, Germany. [5] Lehrstuhl für Biophysik E27, Physik-Department, Technische Universität München, James-Franck-Straße 1, 85748 Garching, Germany. * These authors contributed equally to this work. Correspondence and requests for materials should be addressed to J.B. (email: job.boekhoven@tum.de).

Supramolecular materials consist of molecular building blocks that are assembled via non-covalent interactions such as hydrogen bonding and ionic interactions[1]. Examples include supramolecular polymers[2], bioactive hydrogels[3], photocatalytic fibre that can produce hydrogen[4] as well as macroscopic actuators assembled from small molecules[5]. The past decades have seen tremendous development in the field, and as a result of combined efforts, supramolecular materials are now found in healthcare[6], opto-electronics[7] and others areas. More recent advances have led to supramolecular materials that change their function in response to externally applied stimuli such as light[8], oscillations in pH[9,10] enzymes[11] or even cellular activity[12]. Despite this progress, these materials remain in stark contrast with biological non-equilibrium supramolecular structures especially when it comes to their autonomy and adaptivity[13]. Biological materials typically exist in states far-from-equilibrium, for example, driven by chemical reaction networks that consume chemical fuels like ATP (adenosine triphosphate) or GTP (guanosine triphosphate). As a result, these materials are kinetically controlled, in contrast to man-made materials that are governed by thermodynamic parameters, like free energy landscapes and temperature. This kinetic control endows biological materials with unique properties including a tunable lifetime, robustness, adaptivity and the capacity to self-heal[14]. Inspired by biology, molecular assemblies coupled to chemical reaction networks that involve enzymes have recently been developed to drive man-made molecular assemblers out-of-equilibrium[15,16]. As these systems remain challenging to design, the number of molecular assemblies driven by chemical reaction networks based on exclusively non-biological entities remain limited[17,18], and their use as supramolecular materials with unique properties is largely unexplored terrain.

Here we show a class of autonomously forming and disappearing supramolecular materials. The transient, non-equilibrium materials are produced by a chemical reaction network that can be fully rationalized by a set of kinetic equations. The network comprises mild, man-made reagents and takes place in water making it extremely versatile and scalable. It can be driven by several fuels and coupled to numerous precursors, which allows for the exploration of a wide range of autonomous forming and degrading aqueous supramolecular materials. We demonstrate that these materials are transient, that their lifetimes can be tuned from minutes to hours and that they can be reused by application of another batch of fuel.

## Results

**Characterization of the chemical reaction network**. The chemical reaction network we describe takes place in buffered water at pH 6 at room temperature and consists of a dicarboxylate (precursor) that is converted into an anhydride (product) by consumption of a carbodiimide (fuel; Fig. 1a). The aqueous anhydride product is unstable and rapidly hydrolyses back to the corresponding dicarboxylate. While the dicarboxylate precursor carries two negative charges under the described conditions, the anhydride product is uncharged. We used this hydrophobization, driven by our chemical reaction network, to decrease the solubility of the precursor and thus induce self-assembly. We explored precursors based on aspartate (D) and glutamate (E) because their corresponding cyclic anhydrides were formed with high relative yields. Moreover, being natural amino acids, these precursors were commercially available in numerous variations or could be incorporated in peptide-based supramolecular materials. We tested three fuels and six precursors (Fig. 1b,c). The precursors all bear a relatively large hydrophobic protecting group (fluoren-9-ylmethoxycarbonyl, Fmoc) that aids assembly driven

by π-orbital overlap and hydrophobic collapse. Indeed, we found that the anhydrides of Fmoc-D and Fmoc-E assembled in spherulites and colloids, respectively (vide infra). We introduced the amino acids alanine and valine (A and V, respectively) with a high propensity to form β-sheets with neighboring peptides to increase the anisotropy of the assemblies. Indeed, we found that the anhydrides of Fmoc-AAD, Fmoc-AVD, Fmoc-AAE and Fmoc-AVE assembled into anisotropic fibres (vide infra).

By means of high-pressure liquid chromatography (HPLC) and electrospray ionization mass spectroscopy, the evolution of the concentration of fuel, precursor and product was followed. In all experiments, a batch of carbodiimide fuel was added to a buffered solution of 10 mM precursor. At a concentration of 10 mM, we did not find any evidence of assemblies for any of the precursors (Supplementary Fig. 1a,b). We monitored the pH during a cycle to confirm that under these conditions the buffer has sufficient capacity (Supplementary Fig. 1c). The highest anhydride yields were found for the fuel based on ethyl-3-(3-dimethylaminopropyl) carbodiimide (EDC) as it was relatively stable in our buffer (Supplementary Fig. 1d), yet reactive towards our precursors. Although the other carbodiimide fuels also formed anhydrides and were able to induce self-assembly (vide infra), their yields were drastically lower (Supplementary Fig. 1e). We therefore focused most of our studies on EDC. Each precursor-EDC combination resulted in the transient formation of the anhydride product with a cycle time in the range of tens of minutes to several hours (Fig. 2; Supplementary Figs 2–4). It was found that the chemical reaction networks that used derivatives based on glutamic acid (E) as a precursor were prone to the unwanted N-acylisourea formation as a side reaction, a reaction which yields a stable product of the carbodiimide attached to the precursor. The yields of this unwanted side reaction were not >2% as expressed in amount of fuel added (Supplementary Note 1; Supplementary Table 4), and it is unlikely that these concentrations affect material properties.

A kinetic model was used to describe all relevant reactions in our chemical reaction networks (Supplementary Note 2). These reactions are the direct hydrolysis of EDC, the formation of O-acylisourea by reaction of fuel with precursor, the conversion of the O-acylisourea to the anhydride, the direct hydrolysis of the O-acylisourea to the original precursor, and the hydrolysis of the anhydride to the precursor. The rate-determining step in the anhydride formation was the formation of the O-acylisourea. This reaction had a second-order rate constant which was in the same order of magnitude for each dicarboxylate we tested (Supplementary Table 1). In contrast, the first-order rate constant for anhydride hydrolysis differed up to two orders of magnitude,

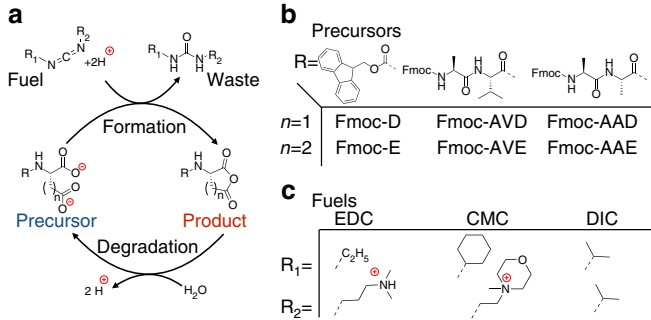

**Figure 1 | Transient anhydride formation by carbodiimide fuels.** (**a**) Scheme of the chemical reaction network for the fuel driven formation of a transient anhydride bond. (**b**) Molecular structures of the dicarboxylates precursors used in this study. (**c**) Molecular structures of the fuels used in this study.

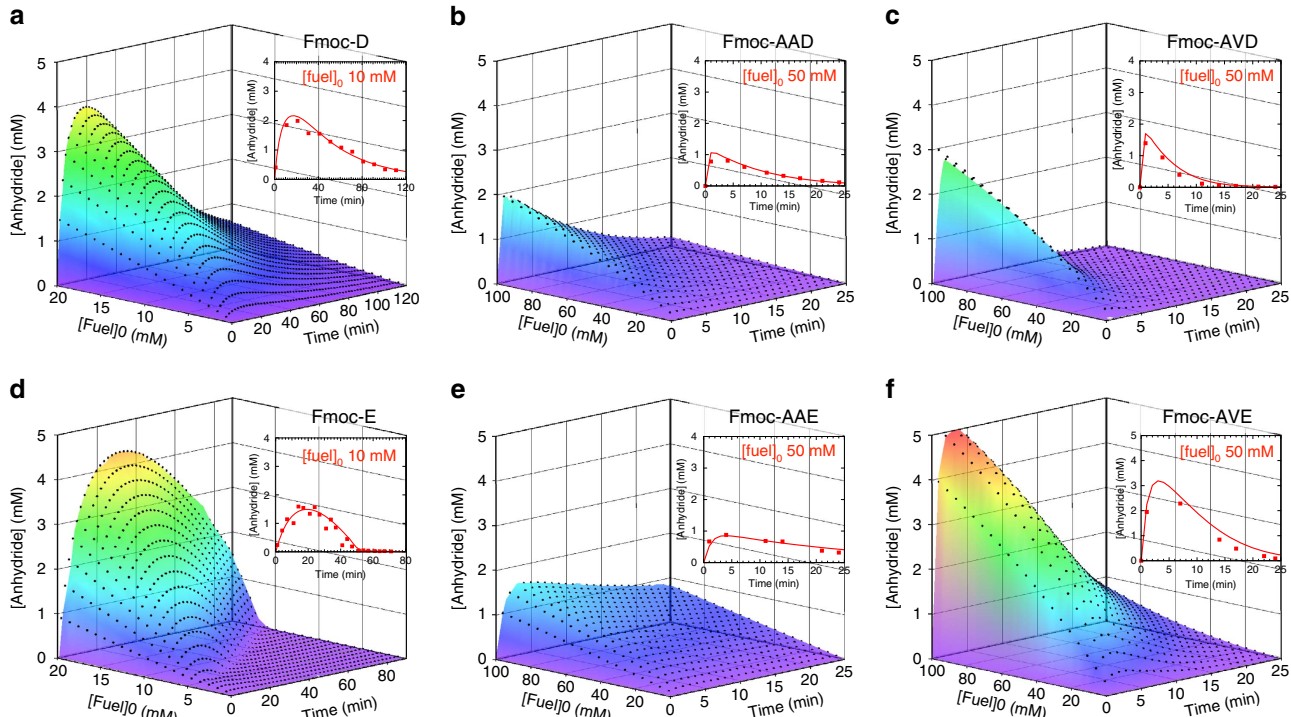

**Figure 2 | Evolution of anhydride concentrations through fuel driven cycles calculated by kinetic models.** (**a**–**f**) 3D plots of the anhydride concentration against time in a dissipative cycle for different initial EDC concentrations for Fmoc-D, Fmoc-AAD, Fmoc-AVD, Fmoc-E, Fmoc-AAE and Fmoc-AVE. The black markers represent the calculated concentration using the kinetic model, the planes represent interpolations between the model data. Note the different *x* (time) and *y* (fuel) axis for **a**,**d** compared to the Fmoc-AA and Fmoc-AV series. The insets show a 2D graph of the concentration anhydride against time for one batch of fuel (10 or 50 mM) as determined by HPLC (markers) and the kinetic model (solid line).

ranging from $3.8 \times 10^{-2}\,\mathrm{s}^{-1}$ for Fmoc-AVD to $3.8 \times 10^{-4}\,\mathrm{s}^{-1}$ for Fmoc-D corresponding to an anhydride half-life of 0.3 and 30 min, respectively (Supplementary Table 1). For all precursors except Fmoc-E and Fmoc-D, the model was straightforward and the rate constants for the above-mentioned reactions were sufficient to fit our data, whether assemblies were present or not (Supplementary Figs 2–4). To our surprise, for Fmoc-E and Fmoc-D, the anhydride hydrolysis rate constant was found to be significantly lower in the presence of assemblies. In effect, the large three dimensional assemblies that these products formed exerted negative feedback on their own hydrolysis rate by protecting the anhydrides from water, thereby increasing the lifetime of the materials (*vide infra*). The feedback was implemented into the kinetic model by introducing a second *k*-value for hydrolysis above a threshold concentration. This threshold concentration was determined such that the model fitted the HPLC data best (see Supplementary Note 2 for a discussion on the feedback and threshold concentration). Using our kinetic models, the evolution of the relevant chemical species through the cycle could be calculated for each precursor (Fig. 2).

**Characterization of self-erasing inks**. We found that application of 10 mM EDC as a fuel to a solution of 10 mM Fmoc-D turned the clear solution into a turbid one, pointing to assemblies formed by the Fmoc-D anhydride with sizes of at least hundreds of nanometers. The assembly process can be explained by the conversion of the two negatively charged carboxylates into a relatively hydrophobic anhydride. The hydrophobization renders the molecules insoluble and induces self-assembly, likely driven by π-orbital overlap and hydrophobic collapse. Confocal microscopy images of these samples revealed the presence of spherulites with diameters of tens of micrometres 10 min after fuel addition

(Fig. 3a). The turbid solutions turned transparent over time and a lack of assemblies was verified by confocal microscopy (Supplementary Fig. 5a–c). The transient turbidity was quantified by UV/Vis spectroscopy and we found that the greater the batch of fuel, the longer the solutions remained turbid (Fig. 3b). An arbitrary threshold of 0.1 absorbance units was chosen to define the turbidity of the samples. We found that 10 mM of fuel kept the samples turbid for $132 \pm 5$ min corresponding to a threshold concentration of $0.3 \pm 0.1$ mM of Fmoc-D anhydride according to the introduced kinetic model. This threshold concentration was used to calculate the lifetime of turbid solutions for other initial fuel concentrations. Indeed, the calculations corresponded well to the experimentally determined lifetimes of turbidity and our active solutions could be kept turbid for 87 to $164 \pm 5$ min for 5 to 20 mM of EDC (Fig. 3c). Finally, the reusability of our system was tested by adding 5 mM of EDC as fuel and measuring the turbidity for 100 min for five cycles. No significant differences between the cycles were observed (Supplementary Fig. 5d).

We sought out the applicability of our the Fmoc-D solutions with transient turbidity as a carrier for temporary messages with a tunable lifetime, that is, as a self-erasing medium (Fig. 3d). Dynamic self-assembled structures have been demonstrated before in the context of self-erasing ink using nanoparticle self-assembly driven by light, which makes tuning the lifetime by amount of energy applied challenging[19]. We immobilized 10 mM Fmoc-D in a polyacrylamide polymer hydrogel and applied high concentrations of fuel as ink with a spray coater through a three-dimensional (3D) printed mask. The transparent gels rapidly turned turbid, but only where the fuel had been applied. Over time the messages auto-erased allowing the reuse of the message carrier. Different EDC concentrations ranging from 0.25 to 2 M were used as ink and all concentrations showed the messages (Supplementary Fig. 5e–h). Using time-lapse photography and

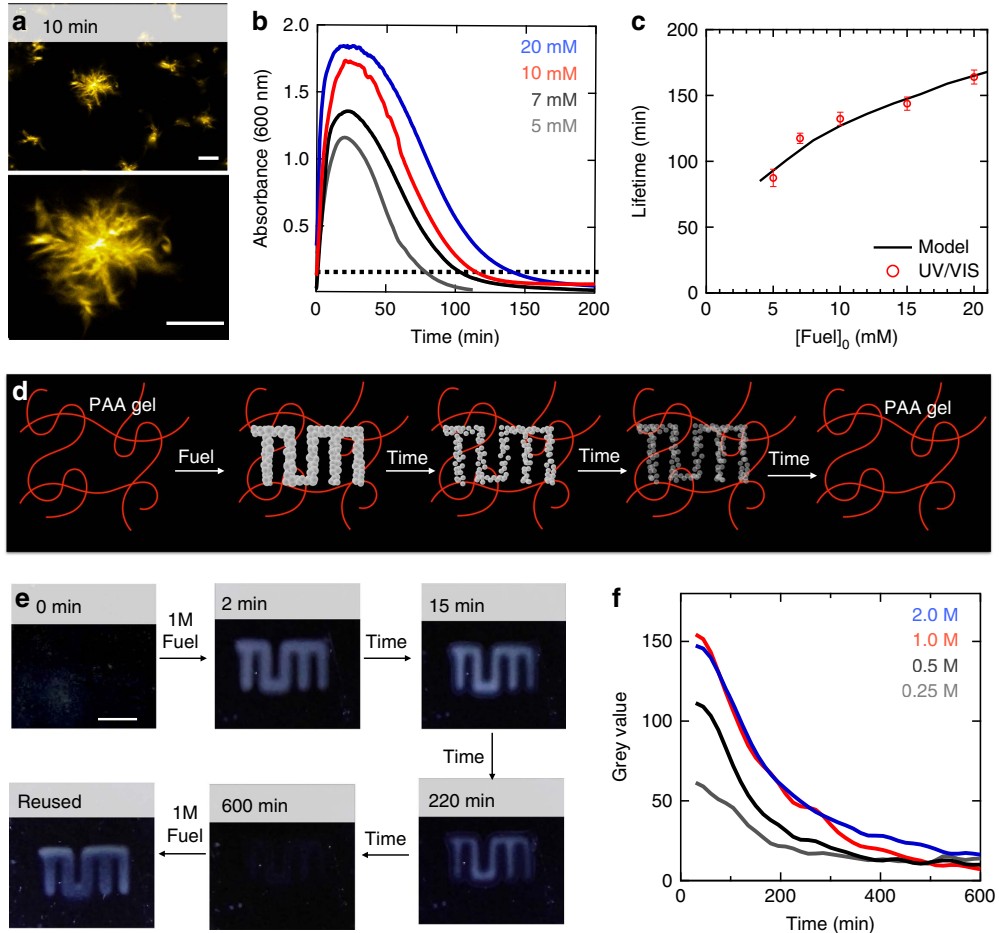

**Figure 3 | Self-erasing inks with a tunable lifetime formed by Fmoc-D.** (**a**) Confocal micrographs of a solution of 10 mM Fmoc-D anhydride 10 min after addition of 10 mM EDC in the presence of 2.5 µM Nile Red (scale bars represent 10 µm). (**b**) Absorbance at 600 nm as a measure for turbidity for solutions of 10 mM Fmoc-D against time after addition of a batch of fuel. The dotted black line represents the selected threshold of 0.1 absorbance units. (**c**) The lifetime of the turbidity of the samples as determined by data from **b** (markers) and calculated by the kinetic model (solid lines) against initial fuel concentration. Error bars represent the s.d. ($n = 3$). (**d**) Schematic representation of self-erasing medium. Fmoc-D is dissolved in a 30% polyacrylamide hydrogel. Only where fuel is applied turbidity appears. (**e**) Photographs of gels over time. The polyacrylamide gels were placed on a black surface and side-lit with white light to improve contrast. After 600 min, no evidence of the original image was found and the material was rinsed and reused (scale bar represents 1 cm). (**f**) Average grey values of photographs of inked areas over time.

image analysis software, we quantified the visibility over time. The initial visibility increased with increasing concentration of fuel, but this trend did not hold beyond 1 M of fuel. At this moment, it is not clear why this trend fails at high fuel concentrations. The visibility of the prints decreased over time and was fully erased after roughly 10 h. Crucially, the lifetimes of the visible prints could be tuned from roughly 200 to 500 min by altering the concentration of fuel (Fig. 3f).

**Colloids with a tunable lifetime.** Next, we studied the behaviour of Fmoc-E in response to fuel. While 10 mM of this precursor yielded clear solutions, application of 5 mM or more EDC formed transient turbid solutions. Confocal microscopy showed, to our surprise, uniform colloids 10 min after application of the fuel (Fig. 4a). Dynamic light scattering (DLS) confirmed the transient presence of particles and showed that their radii initially rapidly grew to a plateau value of 0.6 µm. After the plateau, the particles decreased their size until all had disappeared (Fig. 4b; Supplementary Fig. 6a). This plateau value was independent of the amount of fuel added. Above 20 mM of EDC, the particles precipitated, while below 5 mM EDC no particles were found. Within this window of fuel concentrations, particles were

transiently present. The DLS scattering intensity data was used to find the time where no more particles were found. For instance, in the case of 10 mM of fuel, the scattering intensity reached its original level after $53 \pm 2$ min, corresponding to an Fmoc-E anhydride concentration of $0.07 \pm 0.02$ mM according to the model. This threshold concentration was used in combination with the kinetic model to predict the lifetime of the particles for other initial fuel concentrations. With the model, we found that the lifetime increased linearly from 20 to 120 min for 5 to 20 mM of EDC, and DLS verified that the model predicted these lifetimes correctly. Moreover, application of a second batch of fuel resulted in the reformation of the transient particles with similar radii and lifetimes demonstrating the reusability of these systems (Supplementary Fig. 6b). Even after 15 cycles, the particles still transiently formed with similar radii, albeit with a somewhat shorter lifetime.

For Fmoc-E, the assembly process drastically affected the anhydride hydrolysis kinetics. When small batches of fuel were applied, for instance below 5 mM, no assemblies were formed. In the absence of assemblies, hydrolysis was first order in anhydride concentration. However, in the presence of assemblies, hydrolysis followed 0th order kinetics with a rate constant of

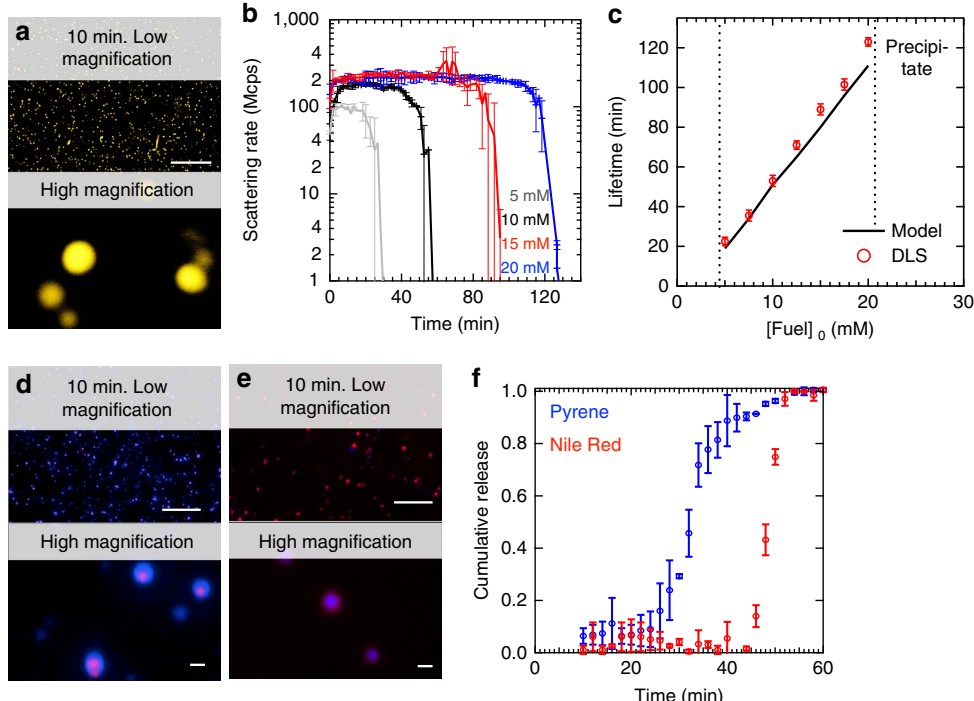

**Figure 4 | Transient colloids formed by Fmoc-E.** (**a**) Confocal micrographs of 10 mM Fmoc-E with 10 mM EDC after 10 min shows the formation of colloids in the presence of 2.5 μM Nile. The scale bars represent 25 μm and 1 μm for the lower and higher magnification, respectively. (**b**) DLS scattering intensity against time for 10 mM Fmoc-E with varying concentration of fuel. (**c**) The lifetimes of the colloids as a function of concentration of initial fuel as determined by DLS and the kinetic model. The two dotted lines indicate the window where particles were found. (**d**) Micrographs of 10 mM Fmoc-E with 10 mM EDC after 10 min that were prepared in the presence of 250 nM Nile Red while 2.5 μM pyrene was added after 5 min. The red (Nile Red) and blue (pyrene) channels were overlaid. Scale bar, 25 (lower magnification); 1 μm (higher magnification). (**e**) Micrographs of 10 mM Fmoc-E with 10 mM EDC after 10 min that were prepared in the presence of 250 nM pyrene while 2.5 μM Nile Red was added after 5 min. The red (Nile Red) and blue (pyrene) channels were overlaid. The scale bars represent 25 μm and 1 μm for the lower and higher magnification, respectively. (**f**) Release profiles of pyrene (blue markers) and Nile Red (red markers) of the particle described in **d**. Data collection was started 10 min after addition of EDC. All error bars represent the s.d. ($n = 2$).

$1.5 \times 10^{-6}\,\mathrm{M\,s^{-1}}$, drastically slowing down the hydrolysis upon self-assembly. In other words, the assembly of the Fmoc-E anhydride into colloids has a negative feedback on its own hydrolysis. Because the Fmoc-E anhydride assembles into large colloids driven by hydrophobic collapse, they likely shield the anhydride from water, effectively excluding it from the chemical reaction network. As a result, anhydride hydrolysis mostly happens on free anhydride in solution and is thus limited by its solubility. The pool of soluble anhydride is replenished by the particles until all particles have disassembled. This finding implies that particles should first collectively grow until the system runs out of fuel and then collectively shrink by surface erosion, that is, degradation from the particle's surface that is limited by the solubility of the anhydride. Such a mechanism contrasts a dynamic mechanism where individual particles grow while others collapse. To confirm the proposed mechanism, we prepared particles in the presence of 250 nM Nile Red. After 5 min, 2.5 μM of the hydrophobic dye pyrene was added, and the particles were imaged after 10 min (Fig. 4d). While Nile Red had mostly incorporated into the core of the particles, pyrene was primarily found in their shell. When the addition of dyes was carried out in the opposite order, the cores had encapsulated Pyrene, while the shell was emitting in the Nile Red channel (Fig. 4e). In contrast, when both dyes were present from the beginning of the cycle, particles were found that had incorporated both dyes homogenously (Supplementary Fig. 6d). These observations confirm the above-proposed mechanism of sequential growth and collapse steps. It is worth to note that also the other two fuels (CMC and DIC) were able to induce the formation of colloids when added to solutions of Fmoc-E (Supplementary Fig. 6d,e).

We used the mechanism of sequential growth and collapse to develop particles that can release hydrophobic agents sequentially. As described above, we prepared particles with Nile Red in their interior and a shell with pyrene incorporated into it. Using a fluorescence spectrophotometer, the release of both dyes from the particles into solution was measured. From the release profiles (Fig. 4f) it can be observed that pyrene release commences after 30 min in the cycle, and most dye has been released after 40 min. In contrast, Nile Red was only released after 50 min. Moreover, when the dyes had Pyrene in their interior and Nile Red in their shell, an opposite release pattern was found (Supplementary Fig. 6f).

**Hydrogels with a predictable lifetime.** When 100 mM EDC was added to solutions of either 10 mM Fmoc-AAD or Fmoc-AAE, each of them rapidly turned into gels. Fmoc-AAD formed weak gels that did not pass the inverted tube test (Supplementary Fig. 7a). Crucially, the gels collapsed within an hour. We confirmed these observations by macrorheology, and found that both precursors formed viscoelastic gels ($G' > G''$) if at least 50 mM EDC was added (Fig. 5a; Supplementary Fig. 7b,c). The maximum gel elasticity for Fmoc-AAD with 100 mM of fuel was obtained after 4 min and was in the range of 100–300 Pa, while the maximum elasticity obtained for Fmoc-AAE was between 40 and 100 Pa and observed after 10 min. Then, the gel stiffness decreased and eventually a solution was found ($G'' > G'$). For 100 mM of fuel, this point occurred at 37 ± 1 and 59 ± 2 min for Fmoc-AAD and Fmoc-AAE, respectively (Fig. 5c). According to our kinetic model, the concentration of anhydride at these times

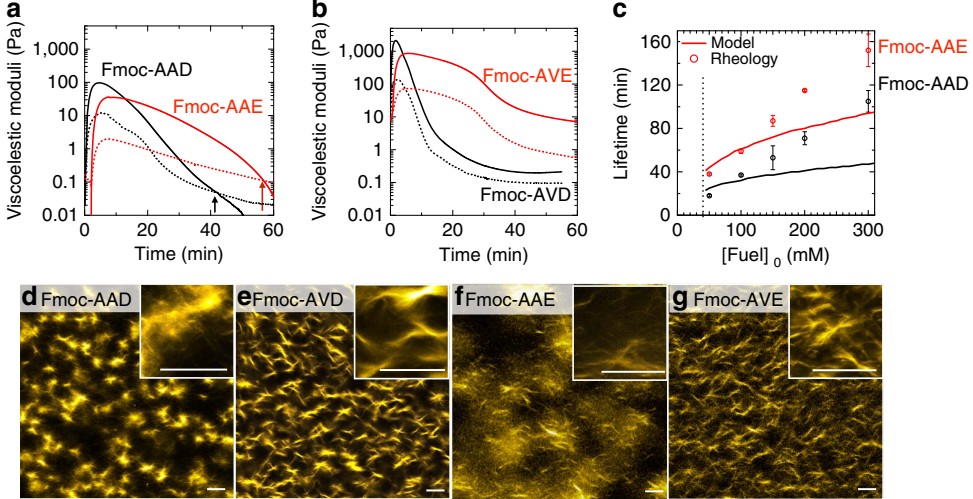

**Figure 5 | Temporary hydrogels formed by Fmoc-tripeptides.** (**a**,**b**) Rheological time sweeps of gels formed by 10 mM Fmoc-AAD and Fmoc-AAE, or Fmoc-AVD and Fmoc-AVE precursor combined with 100 mM EDC. Solid line represents the storage modulus (G'), dashed line represents the loss modulus (G''). (**c**) The lifetimes of the gels described in **a**, as measured by rheology (markers) or calculated by the kinetic model (solid lines). Error bars represent the s.d. ($n = 3$). (**d**–**g**) Confocal microscopy of 100 mM EDC combined with 10 mM Fmoc-AAD, Fmoc-AVD, Fmoc-AAE and Fmoc-AVE after 10 min. Scale bar, 10 μm.

is $100 \pm 50$ and $180 \pm 80 \,\mu\text{M}$ for Fmoc-AAD and Fmoc-AAE, respectively. These concentrations were then used to predict the lifetimes of our gels for other initial concentrations of fuels (Fig. 5c). To our surprise, whereas the prediction made by the model seemed to agree for lower concentrations of fuel, it deviated far from the rheology data for higher concentrations of fuel. For example, for 200 mM of EDC added to Fmoc-AAE, the anhydride concentration falls below 180 μM after 80 min while the gel sustained for roughly 2 h. Although both the experimentally determined lifetime of the gel and the crossing through the threshold concentration of anhydride scaled linearly with the amount of fuel added, their slopes were not equal (Fig. 5c). In other words, we can predict and tune the lifetimes of the transient gels ranging from 18 to 152 min for 50 to 300 mM of fuel, but the moment the gel becomes a liquid does not correspond to one critical threshold concentration of anhydride. These observations could be justified by the complex behaviour of hydrogels that not only rely on the gelator concentration, but also on the type and number of crosslinks, and the length and mechanical properties of the fibres it comprises[20]. To test the reusability of our fuel driven hydrogels, a second batch of fuel was added to the solution after the cycle had finished. Similar gel stiffnesses and lifetimes were found for the first four cycles driven by 50 mM of fuel (Supplementary Fig. 7d).

For both Fmoc-AA precursors, confocal microscopy showed no assemblies before addition of fuel (Supplementary Fig. 8a,c), while addition of 100 mM of fuel rapidly induced the transient formation of fibrillar assemblies (Fig. 5d,f; Supplementary Fig. 8a,c). With this amount of fuel, we found a relatively dense network of fibres for both gelators. Crucially, all fibres of the Fmoc-AA series had completely disappeared after 4 h (Supplementary Fig. 8a,c). We followed the gel formation with circular dichroism spectroscopy at 305 nm, where the signal of the Fmoc-groups is typically recorded[21]. Before initiating the cycle, we found a small positive signal at 305 nm for the Fmoc-AA precursors (Supplementary Fig. 9a). After addition of fuel, the signal rapidly grew, but, over time, returned to its original value pointing to full disassembly after roughly 80 and 200 min for Fmoc-AAD and Fmoc-AAE, respectively. We also followed the Fmoc-fluorescence emission over time during a

cycle as a direct indication for π-orbital overlap between the Fmoc-moieties (Supplementary Fig. 9c–e). Before starting the cycle, a relatively sharp peak at 318 nm was found, indicative for fluorenyl-groups that are not engaged in π–π interactions. Upon addition of fuel, the 318 nm signal rapidly decreased, while a broad peak at 460 nm appeared. Both observations point to π–π stacking of the Fmoc-groups[11]. Over time the 318 nm peak grew back to its original level, while the 460 nm peak disappeared, pointing to loss of the π–π interactions.

**Kinetic trapping of assemblers via dissipative pathways.** When 100 mM of EDC was added to solutions of Fmoc-AVD or Fmoc-AVE, each of them rapidly turned into gels (Supplementary Fig. 7a). To our surprise, these gels did not revert to their original solutions, even after 4 h when no significant amounts of anhydride were present according to our model or HPLC. These surprising observations were confirmed by rheology that showed a decrease in gel-stiffness as the overall anhydride concentration decreased to zero, but never a full recovery to a liquid state (Fig. 5b). The lack of recovery was not a result of the accumulating waste (EDU), as addition of 100 mM or more EDU to solutions of precursor did not change their rheological behaviour (data not shown). Confocal microscopy on the Fmoc-AV systems showed no assemblies before, and dense networks of fibres after addition of fuel (Fig. 5e,g; Supplementary Fig. 8b and d). In contrast to the Fmoc-AA series, the fibres did not disappear, even after 24 h. The longer presence of these fibres allowed us to study them with cryo-transmission electron microscopy (TEM), which confirmed the presence of fibres for both Fmoc-AVD and Fmoc-AVE, with diameters of $6.9 \pm 0.6$ and $4.8 \pm 0.6$ nm, respectively. Finally, circular dichroism showed a small peak at 305 nm before initiating the chemical reaction network that decreased to negative values during the cycle. Although the negative signal decreased in magnitude as the system had run out of fuel, it never returned to the small peak found before initiating the cycle. The original spectrum, as observed before fuel addition, could only be regenerated by heating the sample to 80 degrees and subsequently cooling it down to room temperature (Supplementary Fig. 9a).

We can conclude that all the Fmoc-tripeptides studied in this work assembled into fibres in their anhydride state that subsequently formed viscoelastic gels. The assembly into fibres was driven by β-sheet formation between molecules as evidenced by a Thioflavin-T (THT) assay, and π-orbital overlap between the Fmoc-groups as evidenced by fluorescence-spectroscopy (Supplementary Fig. 9). For the Fmoc-AA series, as the system was running out of fuel, the hydrolyzed anhydrides fibres became unstable which resulted in full disassembly of the fibres and a transition to a liquid. Whether the dicarboxylates left the fibres instantaneously, or a co-assembly of pure anhydride and precursors could exist is not clear at this moment. However, it is evident that when all anhydride had hydrolyzed, all fibres had disappeared. Thus, we conclude that fibres comprising exclusively Fmoc-AAD or Fmoc-AAE dicarboxylates could not exist. In contrast, when all Fmoc-AVE or Fmoc-AVD anhydride had hydrolysed towards their corresponding carboxylates, fibres were still observed. These observations imply that hydrolysis towards the dicarboxylates did not induce (full) disassembly and fibres comprising exclusively the dicarboxylate could exist for the Fmoc-AV series. Only heating and cooling the sample resulted in full disassembly of the dicarboxylate fibres, which demonstrates that the dicarboxylates' thermodynamic minimum is a disassembled state. From the overall results, we conclude that the dissipative cycle, initiated by addition of fuel, kinetically trapped the dicarboxylates into fibres by temporarily converting them into their corresponding anhydrides. The existence of such a kinetically trapped state implies that the molecules did not have sufficient free energy to break out of the fibres, but can only be brought to their thermodynamic minimum by application of heat. The ability to kinetically trap Fmoc-AV, but not Fmoc-AA, likely has to do with the increased β-sheet forming propensity of V versus A[22]. Kinetically trapping molecules in fibres by β-sheet formation and hydrophobic interactions is frequently observed for assemblies in, or close to, equilibrium[23], but are rare in the context of chemically driven out-of-equilibrium cycles[18]. Our observations demonstrate that the balance between assembly and disassembly is crucial in the design of assemblers for materials driven by chemical reactions; when the driving force for assembly is too weak, no assembly will take place, but in contrast, when the driving force is too strong, assemblies can be kinetically trapped in non-favourable states.

## Discussion

We have described a chemical reaction network that drives an anionic dicarboxylate out-of-equilibrium by forming its corresponding metastable non-charged anhydride. Because of the versatility of the network, it can easily be incorporated into peptide-based self-assembling structures, which we showcase with hydrophobic colloids, supramolecular inks and self-assembled hydrogels. Unlike the more classical supramolecular materials, the behaviour of these non-equilibrium materials is dictated by kinetics rather than thermodynamics. This allows us to tune their lifetimes and their decomposition pathway, and we can use them for several cycles. We found that some of the assemblies exert feedback on their own degradation by excluding their building blocks from the chemical reaction network, which we used to our advantage to increase the lifetime of their corresponding materials. Such feedback of assemblies is also one of the crucial ingredients to study more complex life-like behaviour of non-equilibrium assemblies, including robustness, adaptive and oscillatory behaviour of the assemblies.

## Methods

**Materials.** All reagents were purchased from Sigma-Aldrich and Alfa-Aesar and used without any further purification unless otherwise indicated.

**Peptide synthesis and purification.** All peptides were synthesized by solid phase synthesis. Their purity was determined by analytical HPLC as well as electrospray ionization mass spectrometry in positive mode. See Supplementary Methods and Supplementary Tables 2–3 for more details.

**Sample preparation.** Stock solutions of the precursor were prepared by dissolving the peptide in 200 mM MES buffer, after which the pH was adjusted to pH 6.0. Stock solutions of EDC were prepared by dissolving the EDC powder in MQ water and used freshly. Reaction networks were started by addition of the high concentration EDC to the precursor solution.

**Kinetics.** The kinetics of the chemical reaction networks were monitored over time by means of analytical HPLC. A 750 μl sample was prepared as described above and placed into a screw cap HPLC vial. Every 10 min, samples of these solutions were directly injected and all compounds involved were separated and quantified. To avoid aggregation during the injection, samples of Fmoc-D were continuously stirred. In the case of the gelators, the gels were slightly broken manually before injection into the HPLC. See Supplementary Methods and Supplementary Figs 10–15 for more details.

**Kinetic model.** A kinetic model was written using Matlab in which all reactions were described. The concentrations of each reactant were calculated for every 1 s in the cycle and $k$-values were fitted to the HPLC data. See Supplementary Methods for more details.

**UV/Vis.** The UV/Vis measurements were carried out using a UV/VIS spectrophotometer to monitor turbidity. See Supplementary Methods for more details.

**Dynamic light scattering.** DLS measurements on Fmoc-E solutions were performed using a DynaPro NanoStar from Wyatt following a literature procedure. See Supplementary Methods for more details.

**Rheology.** Rheological measurements were carried out on a Antor Paar Modular Compact Rheometer using steel parallel plate-plate geometry. See Supplementary Methods for more details.

**Spray coating set-up.** The fuel deposition via spray coating was performed using a spray gun set up. Applying a constant nitrogen pressure of 2 bar, the flow rate of the solution was set to about 15 μl s$^{-1}$ by tuning the nozzle diameter accordingly. About 300 μl of the respective solution were deposited on the substrate using 2 spray shots of 10 s duration with a short pause. 3D-printed masks were placed onto the substrates in order to create reproducible, detailed images. After the completed spray deposition, the substrates were immediately transferred to an imaging station. See Supplementary Methods for more details.

**Fluorescence spectroscopy.** Fluorescence spectroscopy was performed on a Jasco (Jasco FP-8300) spectrofluorimeter with an external temperature control (Jasco MCB-100). See Supplementary Methods for more details about Fmoc-, THT fluorescence and release assays.

**Cryo-TEM.** Cryo-TEM was performed on a Jeol JEM-1400 plus operating at 120 kV. See Supplementary Methods for more details.

**Circular dichroism spectroscopy.** Circular dichroism measurements were performed on a Jasco (Jasco J750) equipped with a Peltier temperature Control. See Supplementary Methods for more details.

**Confocal microscopy.** Confocal fluorescence microscopy was performed on a Leica SP5 confocal microscope using a × 63 oil immersion objective. Samples were prepared as described above, but with 2.5 μM Nile Red as dye. Twenty microlitre of the sample was deposited on the glass slide and covered with a 12 mm diameter coverslip. Samples were excited with 543 nm laser and imaged at 580–700 nm.

**Data availability.** The data that support the findings of this study are available from the corresponding author upon reasonable request.

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

## Acknowledgements

This work was supported by the Technische Universität München – Institute for Advanced Study, funded by the German Excellence Initiative and the European Union Seventh Framework Programme under grant agreement n° 291763 and the International Research Training Group ATUMS (IRTG 2022). A.R.B., P.M.B. and F.C.L. acknowledge funding by the Nanosystems Initiative Munich (NIM). A.R.B. and O.L. acknowledge funding by Deutsche Forschungsgemeinschaft within the SFB No. 863. MTS acknowledges the European Union's Horizon 2020 Research and Innovation program for the Marie Sklodowska Curie Fellowship under grant agreement n° 747007.

## Author contributions

M.T.S., B.R. and J.B. designed and performed experiments, analysed the data and wrote the manuscript. R.G., F.C.L., C.W., B.K., performed experiments, analysed the data and provided discussion. O.L., A.R.B. and P.M.B. provided discussion. J.B. supervised the research.

## Additional information

**Competing interests:** The authors declare no competing financial interests.

