## [Peer Review File · Nature Communications]

Reviewers' comments:

Reviewer #1 (Remarks to the Author):

This paper by Boekhoven demonstrates a selection of out-of-equilibrium assembly systems based on Fmoc-amino acid and Fmoc-peptide anhydrides that are formed and degraded dynamically using EDC. Different structures can be accessed, kinetics can be described based on kinetic constants of individual processes involved and transient structures (opaque solutions, colloids and gels) are observed. The systems are characterized using rheology, fluorescence labelling, confocal microscopy and those that show longer life times also by CD and TEM.

The paper provides an interesting addition to this rapidly emerging field, with the main areas of novelty: the demonstration of a previously unexplored and versatile reaction of EDC fueled assembly; the morphological tunability observed when using self-assembling Fmoc-peptide anhydrides; and the negative feedback that may be observed when assembled structures are 'protected' against hydrolysis.

The paper is clearly structured, well written and the claims are mostly supported by the data. The paper can be published provided that the following points are addressed.

1. The experimental reaction yields are only reported in SI, it would perhaps be helpful if some of these data were included in Figure 2.
2. During the anhydride formation and hydrolysis a change in pH would be expected. This is not discussed. How does the pH change over time in each of these systems and how does pH impact on the assembly of the precursors?
3. It is well known that Fmoc-peptides and Fmoc-amino acids can form a variety of nanoscale structures. The authors refer to the precursors as un-assembled however the analysis by confocal microscopy clearly precludes the detection of smaller, nanoscopic species. It would be useful if the authors could include TEM (or AFM) images of precursors.
4. No direct evidence of the contribution of aromatic (Fmoc-Fmoc) interactions is provided- time dependent fluorescence spectroscopy would provide evidence of aromatic stacking interactions in precursors and self-assembled structures formed.

Reviewer #2 (Remarks to the Author):

- A. Summary of the key results
- B. Originality and interest: if not novel, please give references
- C. Data & methodology: validity of approach, quality of data, quality of presentation
- D. Appropriate use of statistics and treatment of uncertainties
- E. Conclusions: robustness, validity, reliability
- F. Suggested improvements: experiments, data for possible revision
- G. References: appropriate credit to previous work?
- H. Clarity and context: lucidity of abstract/summary, appropriateness of abstract, introduction and conclusions

A) Summary:

This is a very nice paper from the Boekhoven group in which various FMOC-peptides can be transiently assembled by forming an intramolecular anhydride from a diacid using familiar coupling agents like EDC. The FMOC-peptide-anhydrides are uncharged and therefore less soluble as compared to their charged diacid analogues, which induces assembly. The anhydride reverts back slowly to the diacid leading in most cases to disassembly. In two cases, the structures formed stay kinetically trapped after anhydride ring opening. The various systems were used to create self-

erasing inks, transient colloid formation, and transient hydrogels. Overall this is solid work and a nice contribution to the field.

B) Originality:

The work builds conceptually on the approach of the recent Science paper of Boekhoven, but is sufficiently different to merit publication in NCOMM. This is mainly because the chemistry is very different (EDC anhydride formation here vs. DMS methylation of an acid in Science). The approach here is extended to Fmoc-peptides, which are of interest to a wider audience as compared to the compounds used in the Science paper. Also, EDC is arguably a bit less toxic compared to DMS, but this is still a point for future work. A similar system with complete biocompatibility would be very interesting. I would recommend publication in NCOMM after addressing the points raised in the sections below.

C) Data & methodology:

Overall the methods are solid, but some claims are weak because they are either using non-specified references to the SI, or are not fully supported by experimental data. (note: please add supporting figures in SI and not in the main text).

C1) For example, line 53-54 talks about the hydrolysis rates of the anhydrides, which are "in the range of roughly 20 seconds to 30 minutes (see SI)". The SI is 13 pages, and it took me quite some time to figure out that perhaps Table S1 has columns with constants k_4 and k_5 describing hydrolysis. It is a lot to ask from the reader to put that puzzle together. Please refer to the exact data in the SI, and how one can deduce from that data how your claim in the main text is supported. Can you clarify what is meant exactly with half-life looking at k_4 & k_5 ?

C2) line 69-72: the other fuels (CMC and DIC) are not really studied beyond a single experiment in Fig S1b. I would suggest to remove CMC and DIC from Fig. 1, since they are not really relevant to the main story.

C3) line 75-76: again much is required of the reader; for example, how can I deduce from Figs S2-S4 "whether the assemblies exerted feedback on their chemical reaction network". This would be hard to follow for the average reader. Please combine S2-S4 into a small section in the SI that explains more step-wise how one can see from the data what you write in the main text.

C4) line 78 describes that there are minor impurities (EDC coupled to the peptide itself), which are "not significant to affect material properties (see SI)". Again it is a guessing game. Perhaps SI section 2.4 on rheology could say something about material properties, but I do not find any reference to impurities there. Please be more specific, or soften the claim to be more speculative, e.g., 2% of impurities are unlikely to affect material properties.

C5) line 79: "the symmetric anhydride", to which specific impurity (e.g., use names of side product in SI section 1.3) does this refer?

C6) line 95 speaks about a critical aggregation concentration. Was this measured experimentally or does this come from the model? And where in the SI is the reader supposed to look?

C7) line 116, you could refer to Fig. S5d to make the life of the reader easier.

C8) line 125: please add reference to Fig. S5e-h

C9) line 136: add reference to Fig. S6a. In general, but I think this is clear by this point, please point the reader to where he/she has to look.

C10) lines 157-174: The authors claim that "anhydride hydrolysis only happens on free anhydride in solution". This claim is supported by the fact that the reaction order of hydrolysis goes from 1st

(in absence of colloids) to 0th (with colloids), core-shell growth, and sequential release. I would still suggest to change “only happens on free anhydride” to “mostly happens on free anhydride”. Unless of course the authors can quantitatively show that the rate of hydrolysis changes from 0th to 1st order during a continuously monitored experiment. Also, the authors could use the terminology in degradation of colloidal particles, where people would call this “surface erosion” (cf. https://en.wikipedia.org/wiki/Surface_and_bulk_erosion).

C11) line 184: “liquefaction” cannot be used in this context, since it describes a gas-to-liquid or solid-to-liquid phase transition. I assume the authors mean gel-sol transition or disassembly of the gel. This should be updated throughout the manuscript.

C12) paragraph 241-251: at this point it would be nice to contrast the authors’ finding that the dissipative cycle results in kinetically trapped fibers with the work of ref. 18, where a dissipative cycle was used to trap & free kinetically trapped self-assemblies with strong hydrophobic interactions.

C13) Fig. S8bc: please use clearer legend. Fig. S8e: add units of CD (mdeg?, if so, add path length: 1mm / 1cm?)

D) Statistic: the presented data has appropriate error bars, and supports the main claims. In Fig. S4d the data points for 100 mM seem to be missing (t = 0-13 min.).

E) Conclusions: The conclusions are valid for the results described in the manuscript.

F) Suggestions: I do have a more esoteric point, and that is about the use of “far-from-equilibrium” throughout the manuscript. In my view, which runs along the lines of the founding fathers of non-equilibrium thermodynamics (e.g., Prigogine, De Donder, etc.), “far-from-equilibrium” is reserved for systems where emergent properties (e.g., oscillations, waves, patterns, etc.) are found that are due to instabilities (e.g., highly non-linear chemical kinetics, or reaction–diffusion phenomena) with interactions on length/time-scales far beyond those of the fundamental interactions (typically orders of magnitude beyond the size of the molecules themselves). I would agree that the system of the authors as a whole is a dissipative non-equilibrium system, but the terminology of far-from-equilibrium structures (or assemblies) will increase confusion with reaction-diffusion structures. Clearly, this system is non-equilibrium, but near- vs far-from-equilibrium behavior has not been demonstrated here. I leave this up to the authors, since it is a discussion that is still ongoing in the field of supramolecular chemistry, but “Dissipative non-equilibrium (supramolecular) materials” would be more appropriate than “Far-from-equilibrium (supramolecular) materials”. I am curious to hear the opinion of the authors with respect to this point.

G) References: OK

H) Clarity and context: The manuscript is well written and clear. With the proper references to specific points / figures in the SI this will be easy to read.

Reviewer #3 (Remarks to the Author):

The manuscript describes the transient self-assembly of small peptides into large structures driven by the addition of chemical fuels. Contrary to conventional supramolecular assemblies, these structures have a limited life time which is regulated by the amount of fuel added. The approach followed by the authors is much related to earlier work by the group of Van Esch, but contains enough novel elements to make the manuscript suitable for publication in Nature Communications. The first important issue is the simplicity and general applicability of the system. It is shown that

building blocks can be readily modified to tune both the type of structure formed and its lifetime. Depending on the chemical nature of the building blocks either undefined aggregates, colloids, or gels were formed. The authors then explored the different applications offered by the properties of these structures, i.e. turbidity for self-erasing inks, uptake of dyes for sequential release and hydrogels with controllable lifetimes. Altogether, the results are impressive and highlight the potential of this new kind of chemistry.

Minor issues

- 1) the claim (abstract and page 2, 44) that multiple fuels can be used is not supported by the data. It turns out that only EDC is an effective fuel.
- 2) in Figure 3d, what is the purpose of the fifth image of the gel (it is identical to the fourth) ?
- 3) in Figure 3f, why is no difference observed between 1.0 and 2.0 mM ?
- 4) the sequential release of pyrene and Nile Red (Figure 4e) is one of the highlights of the manuscript? Yet, the results are not entirely convincing as the different release rates could also be reflected by the different polarities of the probes. A control experiment in which first pyrene is added and afterwards Nile Red (leading to an inversion of the release profile) would be extremely convincing.
- 5) in Figure 5 the authors observed a large discrepancy between experimental and calculated values, which is ascribed tentatively to the complex properties of a hydrogel. Isn't it possible that simply the reaction kinetics change in the gel compared to the solution state (in which the separate reactions have been studied).
- 6) the sentence (p4, line 151) 'In the absence of assemblies, hydrolysis proceeded with 1st order kinetics in concentration anhydride' should be changed in 'hydrolysis proceeded with 1st order kinetics' or 'was first order in the anhydride concentration'
- 7) the phrase (p6, line 225) 'Only by heating the sample to 80 degrees and subsequently cooling the original spectrum observed before addition of fuel was regenerated' should be reformulated.

Review report and response from the authors

Reviewer #1 (Remarks to the Author):

This paper by Boekhoven demonstrates a selection of out-of-equilibrium assembly systems based on Fmoc-amino acid and Fmoc-peptide anhydrides that are formed and degraded dynamically using EDC. Different structures can be accessed, kinetics can be described based on kinetic constants of individual processes involved and transient structures (opaque solutions, colloids and gels) are observed. The systems are characterized using rheology, fluorescence labelling, confocal microscopy and those that show longer life times also by CD and TEM.

The paper provides an interesting addition to this rapidly emerging field, with the main areas of novelty: the demonstration of a previously unexplored and versatile reaction of EDC fueled assembly; the morphological tunability observed when using self-assembling Fmoc-peptide anhydrides; and the negative feedback that may be observed when assembled structures are 'protected' against hydrolysis.

The paper is clearly structured, well written and the claims are mostly supported by the data. The paper can be published provided that the following points are addressed.

Authors: We thank the reviewer for her or his time and appreciate the words.

1. The experimental reaction yields are only reported in SI, it would perhaps be helpful if some of these data were included in Figure 2.

Authors: We agree with the referee. We chose not to add all the HPLC data of the chemical reaction cycles in the 3D graphs of Figure 2 as it makes the Figure very crowded. Following the advice of the reviewer, we only added one representative HPLC trace as a 2D graph, and referred to the SI for the rest of the data.

2. During the anhydride formation and hydrolysis a change in pH would be expected. This is not discussed. How does the pH change over time in each of these systems and how does pH impact on the assembly of the precursors?

Authors: The reviewer raises a valid argument. While we carried out pH traces early in this research, we did not mention them in this work. For a concentration of 10 mM precursor with 50 mM of fuel, different MES buffer concentrations were tested (0.02 to 0.5 M) and pH values were monitored over time during the reaction cycle. A concentration of 0.2 M of MES was found to be sufficient for buffering the system at pH 6. At this MES concentration therefore, the pH remains constant during the overall reaction cycle ($\pm 0,1$ pH units).

Alteration in the manuscript: We have added a statement about the pH and added a graph in the SI1.

3. It is well known that Fmoc-peptides and Fmoc-amino acids can form a variety of nanoscale structures. The authors refer to the precursors as un-assembled however the analysis by confocal microscopy clearly precludes the detection of smaller, nanoscopic species. It would be useful if the authors could include TEM (or AFM) images of precursors.

Authors: The reviewer raises a good point. Confocal microscopy is not sufficient to exclude the presence of small (pre)-aggregates and we might be dealing with morphological transitions, rather than "molecularly dissolved to assembly" transitions. However, we don't believe TEM and AFM are valid techniques either. In our experience, drying effects are likely to give rise to some type of nanoscopic structures. For instance, a blank buffer (without precursor) is likely to leave behind deposits of buffer which often show as nanometer sized particles.

We therefore chose to perform a Nile Red assay, that is sensitive to the formation of hydrophobic domains and also sufficiently sensitive to pick up the formation of pre-aggregates. The data shows that 10 mM precursor is sufficiently low to exclude the presence of pre-aggregates. Assemblies start to form at 12 to 15 mM for the Fmoc-AVX tripeptides, around 15 mM for the Fmoc-AAX tripeptides, and was not observed above 15 mM for the Fmoc-E and Fmoc-D. We added the data in the SI.

4. No direct evidence of the contribution of aromatic (Fmoc-Fmoc) interactions is provided- time dependent fluorescence spectroscopy would provide evidence of aromatic stacking interactions in precursors and self-assembled structures formed.

Authors: we agree with the reviewer and performed time resolved fluorescence spectroscopy. The data is in line with what is expected and added to the SI8 and referred to in the main text.

Reviewer #2 (Remarks to the Author):

- A. Summary of the key results
- B. Originality and interest: if not novel, please give references
- C. Data & methodology: validity of approach, quality of data, quality of presentation
- D. Appropriate use of statistics and treatment of uncertainties
- E. Conclusions: robustness, validity, reliability
- F. Suggested improvements: experiments, data for possible revision
- G. References: appropriate credit to previous work?
- H. Clarity and context: lucidity of abstract/summary, appropriateness of abstract, introduction and conclusions

A) Summary:

This is a very nice paper from the Boekhoven group in which various Fmoc-peptides can be transiently assembled by forming an intramolecular anhydride from a diacid using familiar coupling agents like EDC. The Fmoc-peptide-anhydrides are uncharged and therefore less soluble as compared to their charged diacid analogues, which induces assembly. The anhydride reverts back slowly to the diacid leading in most cases to disassembly. In two cases, the structures formed stay kinetically trapped after anhydride ring opening. The various systems were used to create self-erasing inks, transient colloid formation, and transient hydrogels. Overall this is solid work and a nice contribution to the field.

Authors: We thank the reviewer for her or his time and appreciate the words.

B) Originality:

The work builds conceptually on the approach of the recent Science paper of Boekhoven, but is sufficiently different to merit publication in NCOMM. This is mainly because the chemistry is very different (EDC anhydride formation here vs. DMS methylation of an acid in Science). The approach here is extended to Fmoc-peptides, which are of interest to a wider audience as compared to the compounds used in the Science paper. Also, EDC is arguably a bit less toxic compared to DMS, but this is still a point for future work. A similar system with complete biocompatibility would be very interesting. I would recommend publication in NCOMM after addressing the points raised in the sections below.

C) Data & methodology:

Overall the methods are solid, but some claims are weak because they are either using non-specified references to the SI, or are not fully supported by experimental data. (note: please add supporting figures in SI and not in the main text).

Authors: We appreciated the advice and made the references clearer and moved the figures to the SI.

1) For example, line 53-54 talks about the hydrolysis rates of the anhydrides, which are “in the range of roughly 20 seconds to 30 minutes (see SI)”. The SI is 13 pages, and it took me quite some time to figure out that perhaps Table S1 has columns with constants k_4 and k_5 describing hydrolysis. It is a lot to ask from the reader to put that puzzle together. Please refer to the exact data in the SI, and how one can deduct from that data how your claim in the main text is supported. Can you clarify what is meant exactly with half-life looking at k_4 & k_5 ?

Authors: We made clearer where we referred to and explained the calculation of the half-life in table S1.

2) line 69-72: the other fuels (CMC and DIC) are not really studied beyond a single experiment in Fig S1b. I would suggest to remove CMC and DIC from Fig. 1, since they are not really relevant to the main story.

Authors: We would like to emphasize the versatility of the system when it comes to the nature of fuel. We therefore confirmed that CMC and DIC can also induce assembly of Fmoc-E, by addition 50 mM of the fuels to 10 mM of Fmoc-E. In both cases, the solutions turned turbid and colloids were observed by microscopy. We added the microscopy data to Supporting Figure 6 and added a statement in the main text.

3) line 75-76: again much is required of the reader; for example, how can I deduct from Figs S2-S4 “whether the assemblies exerted feedback on their chemical reaction network”. This would be hard to follow for the average reader. Please combine S2-S4 into a small section in the SI that explains more step-wise how one can see from the data what you write in the main text.

Authors: We appreciated the suggestion and the claim is confusing at this point in the text. We explain the negative feedback in detail later, so we altered the text to: “Each precursor-EDC combination resulted in the transient formation of the anhydride product with a cycle time in the range of tens of minutes to several hours (Supplementary Fig. 2-4)”

4) line 78 describes that there are minor impurities (EDC coupled to the peptide itself), which are “not significant to affect material properties (see SI)”. Again it is a guessing game. Perhaps SI section 2.4 on rheology could say something about materials properties, but I do not find any reference to impurities there. Please be more specific, or soften the claim to be more speculative, e.g., 2% of impurities are unlikely to affect materials properties.

Authors: we appreciate the comment and implemented the suggestion in the new manuscript. We referred to the specific section in the SI.

5) line 79: “the symmetric anhydride”, to which specific impurity (e.g., use names of side product in SI section 1.3) does this refer?

Authors: we are referring to the symmetric anhydride that could be expected to form if the carboxylates react intermolecularly, as opposed to intramolecularly. As the statement creates more confusion than clarity, we removed it all together.

The original statement read: "Other side products, including the symmetric anhydride were not observed."

6) line 95 speaks about a critical aggregation concentration. Was this measured experimentally or does this come from the model? And where in the SI is the reader supposed to look?

Authors: The critical aggregation concentration was chosen based on the model and HPLC data. We clarified this in the text and extended our discussion in the SI. We refer the reader to this discussion in the main text.

7) line 116, you could refer to Fig. S5d to make the life of the reader easier.

Authors: we appreciate the comment and implemented the suggestion in the new manuscript.

8) line 125: please add reference to Fig. S5e-h

Authors: we appreciate the comment and implemented the suggestion in the new manuscript.

9) line 136: add reference to Fig. S6a. In general, but I think this is clear by this point, please point the reader to where he/she has to look.

Authors: we referred to the SI in more detail where possible.

10) lines 157-174: The authors claim that "anhydride hydrolysis only happens on free anhydride in solution". This claim is supported by the fact that the reaction order of hydrolysis goes from 1st (in absence of colloids) to 0th (with colloids), core-shell growth, and sequential release. I would still suggest to change "only happens on free anhydride" to "mostly happens on free anhydride". Unless of course the authors can quantitatively show that the rate of hydrolysis changes from 0th to 1st order during a continuously monitored experiment.

Authors: we appreciate the comment and implemented the suggestion in the new manuscript.

Also, the authors could use the terminology in degradation of colloidal particles, where people would call this "surface erosion" (cf. https://en.wikipedia.org/wiki/Surface_and_bulk_erosion).

Authors: we appreciate the new knowledge and implemented the suggestion in the new manuscript.

11) line 184: "liquefaction" cannot be used in this context, since it describes a gas-to-liquid or solid-to-liquid phase transition. I assume the authors mean gel-sol transition or disassembly of the gel. This should be updated throughout the manuscript.

Authors: We implemented the suggestion in the new manuscript. We would like to point out that liquefaction is often used for gel-sol transitions in scientific literature.

12) paragraph 241-251: at this point it would be nice to contrast the authors' finding that the dissipative cycle results in kinetically trapped fibers with the work of ref. 18, where a dissipative cycle was used to trap & free kinetically trapped self-assemblies with strong hydrophobic interactions.

Authors: We agree and added a statement. "Kinetically trapping molecules in fibers by β -sheet formation and hydrophobic interactions is frequently observed for assemblies in, or close to, equilibrium, but are rare in the context of chemically driven out-of-equilibrium cycles

13) Fig. S8bc: please use clearer legend. Fig. S8e: add units of CD (mdeg?, if so, add path length: 1mm / 1cm?)

We separated the spectroscopic assays on the tripeptides in an individual figure (S9), adjusted the caption and added the units. We described the path-length caption and the supplementary information.

D) Statistic: the presented data has appropriate error bars, and supports the main claims.

In Fig. S4d the data points for 100 mM seem to be missing (t = 0-13 min.).

As explained in caption the data is indeed missing for 100 mM of fuel between 3 and 12 minutes. Data acquisition was not possible for these time points as a result of the stiff gels formed.

E) Conclusions: The conclusions are valid for the results described in the manuscript.

F) Suggestions: I do have a more esoteric point, and that is about the use of "far-from-equilibrium" throughout the manuscript. In my view, which runs along the lines of the founding fathers of non-equilibrium thermodynamics (e.g., Prigogine, De Donder, etc.), "far-from-equilibrium" is reserved for systems where emergent properties (e.g., oscillations, waves, patterns, etc.) are found that are due to instabilities (e.g., highly non-linear chemical kinetics, or reaction-diffusion phenomena) with interactions on length/time-scales far beyond those of the fundamental interactions (typically orders of magnitude beyond the size of the molecules themselves). I would agree that the system of the authors as a whole is a dissipative non-equilibrium system, but the terminology of far-from-equilibrium structures (or assemblies) will increase confusion with reaction-diffusion structures. Clearly, this system is non-equilibrium, but near- vs far-from-equilibrium behavior has not been demonstrated here. I leave this up to the authors, since it is a discussion that is still ongoing in the field of supramolecular chemistry, but "Dissipative non-equilibrium (supramolecular) materials" would be more appropriate than "Far-from-equilibrium (supramolecular) materials". I am curious to hear the opinion of the authors with respect to this point.

We thank the reviewer for pointing out the differences. We adjusted the text to "non-equilibrium" where we stated "far-from-equilibrium" before. We are all for clear semantics in a newly developing field.

G) References: OK

H) Clarity and context: The manuscript is well written and clear. With the proper references to specific points / figures in the SI this will be easy to read.

Reviewer #3 (Remarks to the Author):

The manuscript describes the transient self-assembly of small peptides into large structures driven by the addition of chemical fuels. Contrary to conventional supramolecular assemblies, these structures have a limited life time which is regulated by the amount of fuel added. The approach followed by the authors is much related to earlier work by the group of Van Esch, but contains enough novel elements to make the manuscript suitable for publication in Nature Communications. The first important issue is the simplicity and general applicability of the system. It is shown that building blocks can be readily modified to tune both the type of structure formed and its lifetime. Depending of the chemical nature of the building blocks either undefined aggregates, colloids, or gels were formed. The authors then explored the different applications offered by the properties of these structures, i.e. turbidity for self-erasing inks, uptake of dyes for sequential release and hydrogels with controllable lifetimes. Altogether, the results are impressive and highlight the potential of this new kind of chemistry.

Authors: We thank the reviewer for her or his time and appreciate the words.

Minor issues

1) the claim (abstract and page 2, 44) that multiple fuels can be used is not supported by the data. It turns out that only EDC is an effective fuel.

Authors: We would like to emphasize the versatility of the fuel. We therefore confirmed that CMC and DIC can also induce assemblies, by addition 50 mM of the fuels to 10 mM of Fmoc-E. In both cases, the solutions turned turbid and colloids were observed by microscopy. We added the microscopy data to Supporting Figure 4 and added a statement in the text.

2) in Figure 3d, what is the purpose of the fifth image of the gel (it is identical to the fourth)?

Authors: There was no purpose at all. We have adjusted it and thank the reviewer for being attentive.

3) in Figure 3f, why is no difference observed between 1.0 and 2.0 M?

Authors: At this point, we cannot explain the difference. We do know that the signal is not saturated, so it is not a technicaly inherent to the assay. It may be a reaction-diffusion problem. Because we work with such a high concentration fuel, all precursor is converted in the first micrometers and the fuel needs to diffuse further to "find" more precursor.

We added a statement in the main text: "The initial visibility increased with increasing concentration of fuel, but this trend did not hold beyond 1M of fuel. At this moment, it is not clear why this trends fails at high fuel concentrations."

4) the sequential release of pyrene and Nile Red (Figure 4e) is one of the highlights of the manuscript? Yet, the results are not entirely convincing as the different release rates could also be reflected by the different polarities of the probes. A control experiment in which first pyrene is added and afterwards Nile Red (leading to an inversion of the release profile) would be extremely convincing.

Authors: We carried out the proposed experiments and imaged the result with fluorescence microscope (Figure 3f). We also measured the release profile with fluorescence spectroscopy and added that data to the SI (Figure S6).

5) in Figure 5 the authors observed a large discrepancy between experimental and calculated values, which is ascribed tentatively to the complex properties of a hydrogel. Isn't it possible that simply the reactions kinetics change in the gel compared to the solution state (in which the separate reactions have been studied).

Authors: The discrepancy between experimental and calculated values referred to the lifetimes found by rheology and the lifetimes calculated by the model. For Fmoc-D and Fmoc-E, we found that the lifetime could be calculated by the model, by finding a threshold concentration below which no "materials properties" were found. For the building blocks that form gels, this relation was not as simple. We could not find one critical threshold concentration below which we could not find gels by rheology. We attribute that to the complexity of gels. When the G' is lower than G'' , it does not mean no more fibers are present, for instance.

We do not have evidence of changes in rate constants upon self-assembly into fibers. We do see that the model fits the HPLC data very well, even after times that the gels have already transformed into liquids. We believe that that data as such is compelling evidence that transformation to liquids does not change any rate constants.

6) the sentence (p4, line 151) 'In the absence of assemblies, hydrolysis proceeded with 1st order kinetics in concentration anhydride' should be changed in 'hydrolysis proceeded with 1st order kinetics' or 'was first order in the anhydride concentration'

Authors: we appreciate the comment and implemented the suggestion in the new manuscript.

7) the phrase (p6, line 225) 'Only by heating the sample to 80 degrees and subsequently cooling the original spectrum observed before addition of fuel was regenerated' should be reformulated.

Authors: we appreciate the comment and adjusted the text to:

"The original spectrum, as observed before fuel addition, could only be regenerated by heating the sample to 80 degrees and subsequently cooling it down to room temperature."

REVIEWERS' COMMENTS:

Reviewer #1 (Remarks to the Author):

The authors have addressed my comments and the paper can now be accepted for publication.

Reviewer #2 wrote comments to the editor only. They are satisfied with the revisions.

Reviewer #3 (Remarks to the Author):

I strongly recommend publication of the revised version of the manuscript and I would like to congratulate the authors with the impressive results.